## [Decision Letter · Decision Letter 0]

22 Sep 2025

The CD97-PPM1G axis dampens antiviral immunity by dephosphorylating IRF7 in type I interferon pathway

PLOS Pathogens

Dear Dr. Chang,

Thank you for submitting your manuscript to PLOS Pathogens. After careful consideration, we feel that it has merit but does not fully meet PLOS Pathogens's publication criteria as it currently stands. Therefore, we invite you to submit a revised version of the manuscript that addresses the points raised during the review process.

Please submit your revised manuscript within 60 days. If you will need more time than this to complete your revisions, please reply to this message or contact the journal office at plospathogens@plos.org. Please include the following items when submitting your revised manuscript:

We look forward to receiving your revised manuscript.

Kind regards,

Emily Hemann

Guest Editor

PLOS Pathogens

Thomas Hoenen

Section Editor

PLOS Pathogens

Editor-in-Chief

PLOS Pathogens

Michael Malim

Editor-in-Chief

PLOS Pathogens

orcid.org/0000-0002-7699-2064

**Journal Requirements:**

At this stage, the following Authors/Authors require contributions: huasong chang, Wenjing Qi, Rukun Yang, Peili Hou, Ran Kang, Xiaoyu Liu, Yingying Li, Hongmei Wang, and Hongbin He. Please ensure that the full contributions of each author are acknowledged in the "Add/Edit/Remove Authors" section of our submission form.

https://journals.plos.org/plospathogens/s/submission-guidelines#loc-parts-of-a-submission

4) We notice that your supplementary Tables are included in the manuscript file. Please remove them and upload them with the file type 'Supporting Information'. Please ensure that each Supporting Information file has a legend listed in the manuscript after the references list.

5) Please ensure that the funders and grant numbers match between the Financial Disclosure field and the Funding Information tab in your submission form. Note that the funders must be provided in the same order in both places as well.

**Reviewers' Comments:**

Reviewer's Responses to Questions

**Part I - Summary**

Reviewer #1: Here, Chang and Qi et al. seek to broadly extend their previous work on how CD97 regulates antiviral signaling and IFN production. The authors had previously shown that CD97 induces RNF125 leading to ubiquitin-mediated degradation of RIG-I, thereby dampening antiviral responses and promoting the replication of viruses such as VSV (PMID: 37978243). Here the authors find that CD97 also induces and recruits the protein phosphatase PPM1G, which dephosphorylates IRF7 to dampen IFN production. The authors further find that sanguinarine promotes survival of mice infected with IAV H1N1 in a CD97-dependent manner.

Given that the authors had already established the broad function of CD97 in negatively regulating antiviral responses before, most comments will largely focus on the main novel finding here, which is the function of PPM1G in IRF dephosphorylation.

Reviewer #2: In this manuscript, the authors describe a novel mechanism by which CD97 regulates antiviral immunity through the recruitment of PPM1G, a phosphatase that dephosphorylates IRF7, thereby suppressing type I interferon (IFN) responses. CD97, a member of the GPCR family, is identified as a pro-viral factor against multiple viruses. Using biochemical and in vivo approaches in cells and mice, the authors provide evidence that CD97 facilitates viral replication by inhibiting IRF7-mediated IFN responses. Furthermore, they propose sanguinarine (SANG) as a potential therapeutic agent to counteract CD97-mediated pro-viral activity. Overall, the study is carefully executed with appropriate controls, but several points should be addressed to strengthen the manuscript.

Reviewer #3: This study focuses on the interaction between the G protein-coupled receptor CD97 and phosphatase PPM1G, which recruits and dephosphorylates IRF7, leading to the inhibition of its activity. This process blocks the nuclear translocation of IRF7 and subsequent prevent activation of IFN-I signaling. The findings provide a theoretical basis for targeting CD97 as a potential antiviral strategy and developing SANG as a candidate small-molecule antiviral drug. Overall, this study provides evidence of a negative feedback loop of CD97 signaling through restraining IRF7-mediated IFN-I 69 expression in a PPM1G-dependent manner. While the study demonstrates innovation, there are several major and minor issues that need to be addressed.

**Part II – Major Issues: Key Experiments Required for Acceptance**

Reviewer #1: 1) The authors find that PPM1G interacts with both IRF3 and IRF7, but that IRF7 is the primary substrate for its dephosphatase activity. However, in the cell types in which these experiments were performed, IRF3 is the primary IRF driving IFN signaling at early times, while IRF7 may amplify responses after its own induction. At what time point of BEFV infection were the experiments in Fig. 5F performed? This has not been stated in the legend, and IRF7 phosphorylation relative to that of IRF3 is minimal in the -/+ BEFV in Fig 5F. The authors should repeat these experiments with PPM1G overexpression and include early, mid, and late time points of infection to further distinguish between dephosphorylation of IRF3 vs IRF7.

2) What does IRF3 vs IRF7 phosphorylation look like in PPM1G KO cells (sgPPM1G cells from Fig. 4). How does this look like in sgPPM1G cells with CD97 overexpression?

3) IRFs get hyperphosphorylated, but the antibodies used to probe phosphorylation are raised against specific phosphosites. PPM1G may dephosphorylate one or all relevant phosphosites on IRF7/3. Therefore, key experiments from point 2 above should look at IRF7/3 dimers by native PAGE immunoblotting.

4) The authors nicely show that PPM1G catalytic activity is important for IRF7 dephosphorylation/IFN suppression using this D496A mutant. To further prove that PPM1G activity is relevant, and to more convincingly distinguish between IRF7 vs IRF3, the authors should show whether, as expected, IFN induction by the “5D” constitutively active phosphomimetic IRFs is not affected by PPM1G.

5) CD97 both recruits PPM1G and induces its expression through IKZF1. Although CD97 may be internalized under certain circumstances, it is primarily localized to the cell surface. What then is the relevance of PPM1G recruitment by CD97 in its dephosphorylation of IRF7? This should be decoupled by looking at the effect of the CD97 R(819+822)A on downstream PPM1G activity and IFN induction (unless this PRR motif mutant also impacts PPM1G induction).

6) How broad is CD97/PPM1G activity on IRF7 during activation of other innate immune pathways? Is IRF7 activation downstream of TRIF activation affected by PPM1G (and/or CD97)? The authors should test this in TLR3-sufficient cells or by TRIF overexpression.

Reviewer #2: Major concerns

• A large portion of the study relies on BHK21 cells, which are known to be deficient in IFN production and are not physiologically relevant. Is the CD97–PPM1G–IRF7 mechanism specific to BHK21 cells? While the authors partially validate findings in other cells, myeloid cells (where IRF7 plays a central role) were not used and should be considered.

• The analysis of SANG-mediated inhibition of viral replication is underdeveloped. Additional controls are necessary, such as assessing IRF7 activation, PPM1G recruitment, and antiviral gene expression.

• Most biochemical experiments rely on overexpression systems. The authors should validate key findings using endogenous proteins to confirm physiological relevance.

• The connection between CD97 and PPM1G is primarily shown with overexpressed proteins. Stronger evidence is needed to demonstrate endogenous interactions.

• To rule out non-specific effects, NF-κB–dependent genes should be evaluated as additional specificity controls.

Specific Comments

• Figure 1A: Please include molecular weights on western blots. The anti-CD97 blot shows multiple cross-reactive species (lane 2).

• Figure 1B: CD97 expression is significantly decreased at 48 h. Could this be due to proteasomal degradation during viral infection? Some discussion would be helpful.

• Knockdown efficiency appears suboptimal, though downstream effects on viral replication are robust. Please provide quantification of blots. Also, clarify why siCD97-2 was chosen over siCD97-3 in Fig. 1K and L.

• PPM1G has been reported to interact with MAVS, yet this was not observed in the study. The authors should address this discrepancy. Additionally, Fig. 5A (lane 2) requires further justification.

• Figure 5F: The p-IRF7 level appears unexpectedly high in uninfected cells. Please provide a clearer blot showing virus-induced activation.

• Figure 5G: The data on IKKε-mediated IRF3 phosphorylation are unclear and should be improved.

• Figure 5L: Does PPM1G interaction promote CD97 degradation? Please clarify.

Reviewer #3: 1. In Figure 1, the authors conducted BEFV infection experiments in BHK-21 cells following CD97 knockdown. However, the knockdown efficiency at 24 and 48hours post-infection appears suboptimal, as the expression of the internal reference protein was also reduced. The authors did not specify the number of experimental replicates for the Western blot results. Furthermore, it is remarkable that viral replication capacity decreased by approximately 10-fold under this knockdown condition.

2. In this study, the authors utilized CD97 knockout mice for extensive experiments (including those shown in Figure 2, Figure 6, Figure 7, and Figure 8) but failed to provide genotyping or Western blot validation data for these knockout mice. This omission is unacceptable. For every experiment involving CD97 knockout mice, corresponding validation data should be presented in the main or supplementary figures.

3. The study involves extensive infection experiments with Sev, BEFV, and H1N1 viruses. However, viral protein levels were rarely detected when evaluating corresponding indicators. This experimental approach fails to confirm the validity of the experimental system. For example, in Fig. 5F, 5L, 5M, 5N and Fig. 6B, 6C, 6D, 6I, 6J, the absence of viral proteins detection undermines the reliability of the conclusions.

**Part III – Minor Issues: Editorial and Data Presentation Modifications**

Reviewer #1: 1) Fig. 5L needs the no infection condition and total IRF7 blot as appropriate controls.

2) The authors should discuss what may activate CD97 signaling during infection/antiviral immunity.

Reviewer #2: See above

Reviewer #3: 1. The authors used IP-MS to screen for proteins interacting with CD97 and ultimately identified PPM1G as a CD97-interacting protein. The abundance and relevant mass spectrometry data for PPM1G should be clearly presented.

2. The term "ICID50" in line 518 should be corrected to "TCID50", and corresponding errors should be revised accordingly.

3. The transfection dosage of PPM1G in line 580 should specify the amount per unit volume of cell transfection.

4. For all Western blot experiments in the article, the number of independent experimental replicates should be clearly stated. This will allow readers to better evaluate the reproducibility and reliability of the experimental results.

5. PPM1G is primarily localized in the nucleus. In this study, PPM1G interacts with CD97 and subsequently recruits IRF3 for dephosphorylation. According to the author's schematic diagram, this process occurs in the cytoplasm. The authors should address and discuss this apparent discrepancy in the Discussion section.

PLOS authors have the option to publish the peer review history of their article (what does this mean? ). If published, this will include your full peer review and any attached files.

**Do you want your identity to be public for this peer review?** For information about this choice, including consent withdrawal, please see our Privacy Policy .

Reviewer #1: No

Reviewer #2: No

Reviewer #3: No

**Figure resubmission:**

**Reproducibility:**



---

## [Decision Letter · Decision Letter 1]

24 Feb 2026

Dear Dr. He,

We are pleased to inform you that your manuscript 'The CD97-PPM1G axis dampens antiviral immunity by dephosphorylating IRF7 in type I interferon pathway' has been provisionally accepted for publication in PLOS Pathogens.

Best regards,

Emily Hemann

Guest Editor

PLOS Pathogens

Thomas Hoenen

Section Editor

PLOS Pathogens

Sumita Bhaduri-McIntosh

Editor-in-Chief

PLOS Pathogens

orcid.org/0000-0003-2946-9497

Michael Malim

Editor-in-Chief

PLOS Pathogens

orcid.org/0000-0002-7699-2064

The authors have thoroughly addressed reviewer comments through additional experiments and edits to the revised manuscript.

While the endogenous experiment to probe mechanism does not clearly delineate an effect on CD97 vs the CD97-PPM1G complex, the orthogonal methods (overexpression, knockout, biochemical, and endogenous) utilized to address this potential mechanism are sufficient.

Reviewer Comments (if any, and for reference):

Reviewer's Responses to Questions

**Part I - Summary**

Reviewer #1: (No Response)

Reviewer #2: The authors have addressed most of the queries raised. However, some changes would further enhance the quality of the manuscript.

1. Figure 5F: The p-IRF7 levels remain high in uninfected condition.

2. Figure 5G: Elevated p-IRF3 levels are observed in FLAG-IKKε–negative condition.

3. It would be valuable if the authors could further validate the co-IP findings (Figure 5M) using PLA.

4. It is unclear whether SANG suppresses CD97 expression or interferes with the CD97–PPM1G complex. In the Supplementary figure (S5A and B), both PPM1G and CD97 levels appear to be downregulated in the presence of SANG, which makes the mechanism difficult to interpret.

Reviewer #3: In this updated manuscript, the author has done a good job addressing my concerns and questions. I believe the study presents relatively strong evidence demonstrating that CD97 binds PPM1G, which then dephosphorylates IRF7. This blocks IRF7 nuclear translocation and subsequent IFN-I activation, ultimately facilitating viral replication.

**Part II – Major Issues: Key Experiments Required for Acceptance**

Reviewer #1: I am satisfied with the revisions

Reviewer #2: None

Reviewer #3: (No Response)

**Part III – Minor Issues: Editorial and Data Presentation Modifications**

Reviewer #1: (No Response)

Reviewer #2: (No Response)

Reviewer #3: (No Response)

PLOS authors have the option to publish the peer review history of their article (what does this mean? ). If published, this will include your full peer review and any attached files.

**Do you want your identity to be public for this peer review?** For information about this choice, including consent withdrawal, please see our Privacy Policy .

Reviewer #1: No

Reviewer #2: No

Reviewer #3: No

---

## [Editor Report · Acceptance letter]

Dear Mr chang,

We are delighted to inform you that your manuscript, "

The CD97-PPM1G axis dampens antiviral immunity by dephosphorylating IRF7 in type I interferon pathway," has been formally accepted for publication in PLOS Pathogens.

Best regards,

Sumita Bhaduri-McIntosh

Editor-in-Chief

PLOS Pathogens

orcid.org/0000-0003-2946-9497

Michael Malim

Editor-in-Chief

PLOS Pathogens

orcid.org/0000-0002-7699-2064